# Biological Functions of Exopolysaccharides from Lactic Acid Bacteria and Their Potential Benefits for Humans and Farmed Animals

**DOI:** 10.3390/foods11091284

**Published:** 2022-04-28

**Authors:** María Laura Werning, Annel M. Hernández-Alcántara, María Julia Ruiz, Lorena Paola Soto, María Teresa Dueñas, Paloma López, Laureano Sebastián Frizzo

**Affiliations:** 1Laboratory of Food Analysis “Rodolfo Oscar DALLA SANTINA”, Institute of Veterinary Science (ICiVet Litoral), National University of the Litoral-National, Council of Scientific and Technical Research (UNL/CONICET), Esperanza 3080, SF, Argentina; jruiz@vet.unicen.edu.ar (M.J.R.); lpsoto2002@hotmail.com (L.P.S.); lfrizzo@fcv.unl.edu.ar (L.S.F.); 2Department of Microorganisms and Plant Biotechnology, Margarita Salas Center for Biological Research (CIB)-Consejo Superior de Investigaciones Científicas (CSIC), 28040 Madrid, Spain; annel@cib.csic.es (A.M.H.-A.); plg@cib.csic.es (P.L.); 3Department of Animal Health and Preventive Medicine, Faculty of Veterinary Sciences, National University of the Center of the Province of Buenos Aires, Buenos Aires 7000, Argentina; 4Department of Public Health, Faculty of Veterinary Science, Litoral National University, Esperanza 3038, Argentina; 5Department of Applied Chemistry, Faculty of Chemistry, University of the Basque Country (UPV/EHU), 20018 San Sebastián, Spain; mariateresa.duenas@ehu.eus

**Keywords:** exopolysaccharides, EPS, lactic acid bacteria, LAB, probiotics, food-producing animals

## Abstract

Lactic acid bacteria (LAB) synthesize exopolysaccharides (EPS), which are structurally diverse biopolymers with a broad range of technological properties and bioactivities. There is scientific evidence that these polymers have health-promoting properties. Most commercialized probiotic microorganisms for consumption by humans and farmed animals are LAB and some of them are EPS-producers indicating that some of their beneficial properties could be due to these polymers. Probiotic LAB are currently used to improve human health and for the prevention and treatment of specific pathologic conditions. They are also used in food-producing animal husbandry, mainly due to their abilities to promote growth and inhibit pathogens via different mechanisms, among which the production of EPS could be involved. Thus, the aim of this review is to discuss the current knowledge of the characteristics, usage and biological role of EPS from LAB, as well as their postbiotic action in humans and animals, and to predict the future contribution that they could have on the diet of food animals to improve productivity, animal health status and impact on public health.

## 1. Introduction

Lactic acid bacteria (LAB) are Gram-positive, anaerobic aerotolerant, non-spore-forming bacteria with rod or coccus shape. They constitute a group of bacteria commonly found in dairy (fermented), meat and vegetable products, as well as in the gastrointestinal and urogenital tracts of humans and animals, on skin, and in soil and water [1,2]. LAB are used as starter cultures for fermentations and as probiotics in functional foods, and the production of compounds such as nutraceuticals [3,4]. Many LAB can synthesize extracellular polysaccharides (EPS) of high molecular weight. These polymers are strain specific and display great structural diversity. Some EPS remain intimately attached to the surface of the bacteria forming a capsule [5]. However, in this review, the term EPS will refer to those non-capsular polysaccharides that, although they may be loosely associated with the bacterial surface, are largely released into the external environment.

EPS are homopolysaccharides (HoPS) or heteropolysaccharides (HePS), depending on whether their main chain is composed of one or various monomers and their mechanisms of synthesis are, in general, extra- or intra-cellular, respectively [6].

EPS-producing LAB belong to various genera distributed in multiple habitats. The most prominent HePS-producing LAB are *Lactobacillus*, *Lactococcus*, *Streptococcus* and *Enterococcus* strains frequently isolated from fermented dairy products and the human gastrointestinal tract (GIT) and feces, whereas the majority of HoPS-producing LAB are *Lactobacillus*, *Leuconostoc*, *Pediococcus*, *Streptococcus*, and *Weissella* strains isolated from animal GIT, vegetables and fermented beverages [7]. The wide distribution of EPS production indicates that it must confer some competitive advantage in the ecological niches of these bacteria. Indeed, some possible physiological roles proposed for LAB-EPS are the protection of the bacteria from stress conditions, including environmental pH, osmotic stress or desiccation, and from bacteriophages, antibiotics, and lysozymes, enabling LAB persistence in various niches including biofilm formation [8,9]. In general, synthesis of EPS is not involved in the generation of energy, neither are EPS used as a carbon source by the producing microorganism [10]. Nevertheless, synthesis of the HoPS dextran, by *Weissella* and *Leuconostoc* strains, as well as lactobacilli, is accompanied by the hydrolysis of sucrose generating monosaccharides which are used by these LAB as a carbon source [11].

Several applications for EPS from LAB have been proposed. With the exception of HoPS dextran, until now, only the in situ application of EPS-producing LAB has been economically viable, for example, as starter cultures, instead of purified EPS due to the insufficient levels normally produced, as well as production costs. Particularly in the fermented food industry, EPS production by LAB contributes to improving the organoleptic quality, sensory and rheological properties, as well as stability, of the final products. Moreover, by their addition to these food products, EPS can be considered as functional postbiotic ingredients due to their postulated human health benefits, such as immuno-modulation, anti-oxidative, anti-inflammatory, anti-microbial, anti-tumoral, or cholesterol-lowering properties, or as microbiome modulators [12,13,14]. Indeed, many probiotic bacteria (or bacteria with potential as such), are producers of a wide range of EPS and the scientific evidence suggests that some of the health effects attributed to probiotic LAB may be due to their production of the polymers. Thus, bacterial resistance to gastrointestinal stress and the persistence of the bacteria in the gut ecosystem are influenced by their EPS production [9].

The present work includes an updated review of general issues related to LAB-EPS including its classification, biosynthesis and production and especially its current and potential applications. However, the main objective of this review is to detail the current knowledge of EPS production by probiotic LAB. The role of LAB-EPS is reviewed in detail, both in terms of the probiotic properties and the biological activities attributed to their producing bacteria and whose effects can be both local and systemic in the host. Finally, the authors consider the potentially beneficial future contribution they could make by inclusion in the diet of farmed animals for the improvement of productivity, sanitation and public health.

## 2. Structure, Production and Purification of EPS

### 2.1. Structure

LAB produce a wide diversity of EPS as defined by their monomer composition, molecular mass and structure. According to their monosaccharide composition and biosynthetic pathway, they are classified as HePS or HoPS. HePS possess variable molecular masses (generally up to 10^6^ Da) with two or more different types of monosaccharides organized in repeating units with a variable number from tri- to octa-saccharides. The monomers, frequently D-glucose, D-galactose, and L-rhamnose are usually joined by β-(1,4) or β-(1,3) and α-(1,2) or α-(1,6) linkages, but other components, such as L-fucose, D-mannose, N-acetylmonosaccharides, D-glucuronic acid or glycerol may also be present [6]. In addition, they can be modified with pyruvate and phosphate. Out of the 81 reported structures, a monosaccharide occurrence in 55 unique repeating units has been described; among them, 40 correspond to *Lactobacillus*, 11 to *Streptococcus* and 4 to *Lactococcus* genera [15]. The large variation in composition, linkage type and branching patterns leads to a large diversity in the polymer structures and sizes, resulting in diverse technological functions.

### 2.2. Production

The HePS are synthetized by intracellular glycosyltransferases using sugar nucleotides as substrates. These enzymes are encoded by genes located in the *eps* gene cluster, together with other genes encoding proteins involved in chain-length, polymerization and export of the EPS, as well as in the regulation of gene expression [5,6]. Although the exact mechanisms are not fully understood, the role of the molecular determinants of LAB-EPS biosynthesis has been recently reviewed [7,9,15].

With respect to HoPS, these polysaccharides are widely produced by LAB, being reported in species belonging to *Weissella*, *Streptococcus*, *Pediococcus*, *Oenococcus*, *Lactobacillus* and *Leuconostoc* genera; their production has been extensively reviewed in [8,12]. In general, these polymers display high molecular masses (up to 10^8^ Da) and are composed of glucose (glucans), fructose (fructans) or galactose (polygalactans). These polymers are subdivided based on the type of glycosidic linkage, and the type and degree of branching. Glucans include α-glucans (e.g., dextran, mutan, alternan and reuteran) and β-glucans, whereas there are two classes of fructans, levan-type and inulin-type, both being β-fructans. Extracellular glycansucrases, encoded by a single gene, catalyse the synthesis of the majority of these HoPS using sucrose as a substrate.

However, other mechanisms of glucan biosynthesis have been reported. Thus, β-glucans are synthesized by a single transmembrane glycosyltransferase that utilizes UDP-glucose as the donor substrate [16]. Furthermore, Mayer et al. [17] have recently reported that a putative bactoprenol glycosyltransferase and flippase are essential to branched dextran HoPS biosynthesis in *L. johnsonii* F19785.

Both, the yield and molecular mass of the EPS produced by LAB can be affected by intrinsic, as well as environmental factors, including medium composition, growth conditions and incubation time. HePS are synthesized in low amounts (20 to 600 mg/L), although, under optimal culture conditions, a few strains produce higher amounts, e.g., *Streptococcus thermophilus* ASCC 1275 and *Lacticaseibacillus rhamnosus* RW-9595M (formerly *Lactobacillus rhamnosus*) producing 1 g/L, and 2.7 g/L, respectively. However, production of HoPS is high, with some strains synthesizing even 10 g/L, e.g., *Limosilactobacillus reuteri* Lb121 (formerly *Lactobacillus reuteri)*, which produces an α-glucan and a β-fructan. The difference in the production of both classes of polymers may be because the synthesis of the HePS, compared to that of the HoPS, is a more complex mechanism related to the central metabolism of the bacterium. In addition, the isolation and purification procedures can lead to significant differences in the polysaccharide concentration measured [6,15].

For the technological application of EPS from LAB, an optimal and sufficient production, both in situ and ex situ, is imperative, not only to improve their yield, but also to obtain a particular functionality. In this respect, metabolic engineering could improve EPS production, although it might be of limited value due to the inherent complexity of the synthetic mechanisms, especially in the case of HePS [18]. Nevertheless, an example of this was the finding that NADH oxidase overexpression had a significant effect on EPS production in *Lacticaseibacillus casei* LC2W (formerly *Lactobacillus casei)* [18].

Another strategy is the optimization of the production conditions, as well as the regulation mechanisms. Environmental stress can improve the production of EPS by LAB and produce customized EPS with desired functionality. For example, EPS synthesis by *L. helveticus* ATCC 15807 is stimulated under acidic pH stress and inhibited under sodium chloride osmotic stress [19]. Moreover, in *L. helveticus* 6E8, at high sugar concentrations, there was a shift from bacterial growth to EPS synthesis and secretion [20].

The study of the conditions and mechanisms of expression regulation of dextran production in *Leuconostoc lactis* AV1n revealed that this strain synthesizes the HoPS efficiently at low temperature; therefore, this LAB could be a good candidate for the in situ production of the polymer during the manufacture of some functional foods (e.g., kefir) [11].

### 2.3. Purification

Both classes of polysaccharides can often be purified by similar procedures. However, to achieve the best possible yield, it is necessary to establish a combination of conditions and procedures for each case [21]. Laboratory methods for the production, isolation, purification and quantification of EPS were recently reviewed providing a detailed description of this topic [18,22]. Briefly, EPS can typically be recovered from bacterial culture supernatants using variations of a general method, including precipitation from the broth, and purification of the polymer from the precipitate. Depending on the medium, the first step is often protein elimination using trichloroacetic acid or proteases. After centrifugation, the biopolymers are recovered from the growth medium by precipitation with cold ethanol or acetone. The EPS is then dissolved in distilled water and dialyzed in a bag with a 12–14-kDa cut-off to eliminate low-molecular-mass carbohydrates, and, subsequently, the EPS are lyophilized. Additional chromatographic purification steps are usually required if the structure of the EPS is to be characterized and potential biological applications are to be explored.

The structures of EPS have been extensively investigated; generally, their study includes determination of molecular weight by gel permeation chromatography and asymmetric field-flow fractionation (AF4) or *by size exclusion* chromatography coupled to multiple-angle laser light scattering, the composition of monosaccharides by liquid-gas chromatography followed by mass spectrometry, and the monosaccharide pattern by a combination of methylation and nuclear magnetic resonance analysis. Finally, the microstructures of the EPS are observed by scanning electron microscopy or atomic force microscopy [7].

The chemical structures and molecular masses are properties that have been correlated with the technological characteristics and biological functions of the EPS in diverse applications. A comprehensive review of the relationship between the structure and function of EPS from LAB and bifidobacteria has recently been reported [7,23]. However, in the following sections, the postbiotic effects known to date of EPS from LAB will be discussed, in particular, as well as their role in the probiotic properties of the producing LAB.

## 3. Biological Functions of the EPS

### 3.1. Antimicrobial Effects

The antimicrobial activity of various EPS from LAB against a wide variety of pathogenic microbes has been investigated using both in vitro and in vivo assays [24,25,26]. EPS can exert their antimicrobial action indirectly, either (i) through the stimulation of the innate and adaptive immune response, or (ii) by promoting the growth and/or formation of biofilms of other beneficial commensal bacteria or probiotics. Although these topics are discussed in the immunomodulatory effects and prebiotic properties sections, respectively, some examples of EPS action against pathogens through both modes of action are described below. Dextran produced by *L. mesenteroides* NTM048 stimulated mucosal IgA secretion. Using an in vitro assay, NTM048 EPS stimulated T helper (Th1) and (Th2) cells-mediated responses, as well as total and antigen-specific IgA production in splenocytes [27]. In addition, it has been demonstrated in vivo that the HePS from *L. fermentum* UCO-979C is partially responsible for increased resistance against *Helicobacter pylori* infection by modulating the gastric innate immune response [28].

The antimicrobial activity of LAB-EPS could also be explained in vivo by: (i) the prebiotic effect that helps LAB in gut colonization, (ii) the protection of commensal microorganisms from the adaptive immune response in the host, and (iii) the enhancement of their competition with pathogenic bacteria [29]. Although there are no related reports in LAB, this was demonstrated for the HePS from *Bifidobacterium breve* UCC2003 [30] and HoPS from *Bacillus subtilis* HMNig-2 [31] in murine models.

These polymers can also directly promote antimicrobial activity in the following ways: (i) inhibiting the growth of pathogenic microorganisms; (ii) interfering with their adhesion to the intestinal epithelium; and (iii) by preventing or reducing the formation of biofilms by pathogenic bacteria. The anti-biofilm properties of LAB-EPS are examined below; some examples are briefly mentioned. HePS producing *L. rhamnosus* and HePS from *L. plantarum* WLPL04 exhibited strong inhibition against biofilm formation by pathogenic bacteria, including *Salmonella enterica* serovar *typhimurium* [32], as well as *Pseudomonas aeruginosa*, *Escherichia coli* O157:H7, and *Staphylococcus aureus* [33]. Moreover, the disruptive effect of EPS produced by *Loigolactobacillus coryniformis* NA-3 (formerly *Lactobacillus coryniformis*) on pre-formed *S. typhimurium* and *Bacillus cereus* biofilms was the greatest of those tested and was reported to be 80 and 90%, respectively [7]. A possible mechanism of antimicrobial action is that the EPS of LAB interact with the signaling molecules involved in the formation of biofilms on the surface of these pathogens, interrupting their formation and thus exerting an antimicrobial effect [34].

With respect to the ability of EPS to inhibit pathogenic microorganisms, it was reported that HePS producing *L. rhamnosus* isolated from human breast milk showed, in an in vitro assay, strong anti-bacterial activity against pathogenic *E. coli* and *S. typhimurium*, [32]. Similarly, HePS produced by *L. gasseri* [35] and *L. kefiranofaciens* DN1 [36] have shown in vitro antibacterial activity against several foodborne pathogens, such as *Listeria monocytogenes* and *Salmonella enteritidis*. Moreover, in a recent study, HePS extracted from *L. plajomi* PW-7 (formerly *Lactobacillus plajomi*) showed antibacterial activity against *H. pylori*, *S. aureus*, and *E. coli* [37]. With respect to the potential inhibitory mechanism of pathogenic bacteria by EPS, it was proposed that this can be effected by disrupting the integrity of membranes and releasing the contents of soluble proteins [38]. Indeed, the disruption of the cell membranes of some Gram-positive and Gram-negative pathogenic bacteria by EPS from *L. plajomi* PW-7 was confirmed by electron microscopy [37]. Another mechanism, proposed by Salachna et al. [39], is that the EPS may promote the accumulation of secondary metabolites in the growth media, which might adversely affect both Gram-positive and Gram-negative pathogens [39].

The interaction of the bacterial EPS kefir with bacterial or eukaryotic cells, suggests that the antimicrobial action of this EPS occurs via the blockade of receptors or channels of the outer membrane [40].

Antifungal activity of LAB-EPS has been reported [24]. The activity of EPS-producing *L. rhamnosus* GG, its non-producing mutant as well as purified EPS, revealed that GG EPS might be involved in reducing hyphal formation and decreasing Candida adhesion through: (i) co-aggregation, (ii) immunomodulation of the host epithelial cells, and (iii) competition for binding sites [41].

Furthermore, it has been reported that EPS promoted antiviral activities by immunomodulating native and adaptive response and by interfering with the adhesion and /or reproduction of viral particles (antiviral properties linked with immune system responses are discussed in the antiviral effects section). On the other hand, some studies suggest that EPS can act directly against various viral pathogens [24]. With respect to this effect or mode of action, it was shown that EPS 26a from *Lactobacillus* spp. obstructed adenovirus type-5 (HAdV-5) reproduction [42] and EPS from *L. plantarum* LRCC5310 interfered with rotavirus attachment to cells in vitro [43]. In addition, tests in young mice, with EPS-LRCC5310, produced a reduction in rotavirus replication in the intestine and in the duration of diarrhea, with consequent reduction in the recovery time of the mice [43].

The interference activity exerted by LAB-EPS on pathogens in the host can be assessed through the ability to co-aggregate and decrease the accessibility of pathogens to the intestinal epithelium. Strains of *L. delbrueckii* subsp. *bulgaricus* can co-aggregate with *E. coli* [44]. This mode of action has also been demonstrated by utilization of intestinal cell line models. *L. paraplantarum* BGCG11, a producer of HePs, but not its isogenic non-producer strain, showed a protective effect on mucous membranes by hindering the contact of *E. coli*, *L. monocytogenes* with epithelial cells HT29-MTX [45]. In another study, it was determined that the HePS produced by the strain *L. paracasei* subsp. *paracasei* BGSJ2-8 was involved in adhesion to epithelial intestinal cells and decreased the *E. coli* association to Caco-2 cells [46]. In addition, HePS-producing *L. plantarum* WLPL04 isolated from human breast milk had a significant inhibitory effect on the adhesion of *E. coli* O157:H7 to human intestinal epithelial cells [47]. A posteriori purified HePS-WLPL04 had a similar inhibitory effect on the adhesion of *E. coli* O157:H7 to HT-29 cells in competition, replacement, and inhibition assays [33]. In addition, in an in vivo study, *L. johnsonii* FI9785, a HePS and α-glucan producer, was effective in suppressing the colonization and persistence of *Clostridium perfringens* in poultry and reduced colonization by *E. coli* of the small intestine [48,49]. Through hydrophobicity and autoaggregation, EPS synthesized by *L. johnsonii* FI9785 competitively inhibited these pathogens [50].

EPS can also exert an antagonistic effect on the inflammatory response caused by pathogenic microorganisms in the intestinal epithelium through modulation of the immune system. Porcine intestinal epithelial cell lines have been used to determine the antagonistic effects of EPS-producing strains, such as *L. delbrueckii* TUA4408L, by attenuating the inflammatory response induced by enterotoxigenic *E. coli* [51].

The antimicrobial action of EPS depends on their composition and structure. The molecular weight, composition, and charged groups are reported to be particularly related to this activity. Purified EPS fractions of high molecular size showed stronger antibacterial effects against Gram-negative bacteria, whereas the opposite trend was observed for Gram-positive bacteria [52]. Moreover, EPS composition is implicated in the potential interaction with pathogens. In this respect, analysis of the EPS-deficient mutant *L. paracasei* subsp. *paracasei* BGSJ2-8, which produces an EPS with a different composition from that of the wild-type strain, indicated that the wild-type EPS is essential to reduce *E. coli* association with Caco-2 cells [46]. Moreover, in a study that evaluated the antibacterial effect of *L. plajomi* PW-7-EPS components, it was found that galactose had a greater effect on *E. coli*, while glucuronic acid had a greater effect on *S. aureus* and xylose had a stronger effect on *H. pylori* [37].

Finally, substitutional modifications of EPS, including sulfonation, phosphorylation and acetylation have been reported to affect their antimicrobial activity. The sulfation of EPS leads to potent antiviral activity. The modification was shown to be effective in inhibiting virus-cell interaction. In addition, this activity was demonstrated, among others, for hepatitis B, herpes and influenza viruses [53]. Sulfated EPS from *L. plantarum* ZDY2013 and *S. thermophilus* exhibited a greater antimicrobial effect against various Gram-positive and harmful pathogens than non-sulfated compounds [54,55].

### 3.2. Anti-Biofilm Properties

Biofilms are extracellular matrices which adhere to surfaces, composed of a complex of nucleic acids, proteins, polysaccharides and lipids. Many bacterial species, including pathogenic bacteria, become more resistant to extracellular stress conditions by producing biofilms. In several pathogenic microorganisms, environmental stress can trigger the formation of biofilms, which increase adhesion and protection against the host response. Consequently, biofilms play an important role in pathogenesis [56]. Colonization of the chicken gastrointestinal tract and oviducts by *S. typhimurium* is largely due to its ability to adhere and to form biofilms. In chicken epithelial cell lines (Hep-2), it has been shown that certain EPS contribute to biofilm formation [57].

There is increasing scientific evidence supporting the position that various EPS extracted from LAB can reduce or inhibit microbial biofilms and that, therefore, they have potential application in the design of new strategies to deal with bacterial biofilm-associated infections and food safety issues [24]. It has been shown that many EPS of lactobacilli can intervene in the formation of biofilms or disperse those already formed by pathogens. EPS produced by *L. acidophilus* have been demonstrated to inhibit the biofilm formation of a number of pathogens, including enterohemorrhagic *E. coli* and *S. enteritidis* [58]. Previous studies have demonstrated that the EPS from *L. plantarum* YW32 and *L. acidophilus* A4 were found to possess anti-biofilm activity against both Gram-negative and Gram-positive pathogens [59,60]. In addition, HePS from *L. fermentum* LB-69 [61] and *L. gasseri* FR4 [35] displayed their highest biofilm inhibition on *B. cereus* RSKK 863 and *L. monocytogenes* respectively.

Anti-biofilm activity was also reported for other LAB. Dextran produced by *W. confusa* has been shown to have antibiofilm activity against *Candida albicans* SC5314 [62]. *L. citreum* isolated from beef sausages produced dextran and levan. These polysaccharides showed high activity in disrupting pre-formed biofilms and biofilm inhibition [63].

The quorum sensing (QS) system consists of a mechanism in which bacteria produce molecules (generally oligopeptides in Gram-positive bacteria and acyl-homoserine lactone in Gram-negative bacteria) by means of which they detect the size or density of the other bacteria that surround them and thus regulate the formation of the biofilm [29]. It has been suggested that EPS can either modify the bacterial coat and thereby hinder the attachment of bacteria to surfaces, or act as a signaling molecule, and regulate gene expression involved in biofilm formation [64]. In this sense, as mentioned in the previous section, several sulfated EPS exhibited a stronger inhibitory effect for multiple Gram-positive and negative pathogens than those free of sulfate. A possible reason for this is the interruption of the signals that mediate biofilm formation or the efflux pathway of water-soluble proteins with damage to the cell membrane [29].

### 3.3. Prebiotic Properties

The term prebiotic was recently defined as a “substrate that is selectively utilized by host microorganisms conferring a health benefit” [65]. Orally delivered prebiotics are “non-digestible food ingredients through the gastric and small intestine, and when they reach the large intestine in humans, they are selectively utilized by some intestinal bacteria populations, stimulating the growth and/or activity of intestinal microbiota and thereby with beneficial effect on host health” [66]. They are generally poly- and oligosaccharides, such as inulin (β-(2→1)-fructan, galacto-oligosaccharides and fructo-oligosaccharides. The metabolism of these carbohydrates generates products, such as short-chain fatty acids, gases, and organic acids, with positive effects on the host, including to: (i) provide energy to intestinal colonocytes, (ii) inhibit pathogenic bacteria, and (iii) modulate the human or animal metabolism [6].

EPS synthesized by LAB have a potential role as prebiotics as they are resistant to GIT digestion and can be selectively metabolized by beneficial gut bacteria, especially *Bifidobacterium* spp. and *Lactobacillus* spp. [67]. The majority of LAB-EPS with demonstrated prebiotic potential are HoPS and their associated oligosaccharides. It is possible that the more complex composition of HePS is responsible for the low capability of the gut microbiota to hydrolyze and further metabolize this type of polymers, limiting their prebiotic potential. Supporting this hypothesis, it has been demonstrated that the simple primary structure of HoPS allows fecal microbiota to ferment them [68]. Dextran from *W. cibaria* RBA12 showed high resistance to hydrolysis by artificial gastric juice, α-amylase and intestinal fluid and enhanced the growth of probiotic *Bifidobacteria* spp. and *Lactobacillus* spp. [69]. Levan from *F. sanfranciscensis* had a bifidogenic effect and modified the gut microbiota composition [70]. In addition, a purified β-glucan synthesized by *P. parvulus* 2.6 R, isolated from ropy cider [71,72], improved the growth of three probiotic strains *L. plantarum* WCFS1, *L. acidophilus* NCFM, and *L. plantarum* WCFS1 β-gal, which over-express a β-glycosidase enzyme [73]. The prebiotic effect of a linear dextran produced by an *L. pseudomesenteroides* strain has recently been investigated by feeding mice with the polymer [74]. The results revealed that the HoPS affected the structure of the gut microbiota of the treated mice, by decreasing the ratio of Bacteroidetes to Firmicutes [74].

Furthermore, some HePS from LAB which are mainly composed of glucose, mannose, galactose and rhamnose, have also shown prebiotic activity. A mannose-rich HePS produced by *L. rhamnosus* GD-11 also stimulated the growth of bifidobacteria. Both HePS fractions (r-EPS1 and r-EPS2) produced by *L. delbrueckii* subsp. *bulgaricus* SRFM-1 promoted bifidobacteria metabolism in a human fecal sample, generating a high concentration of short-chain fatty acids. Both EPS-fractions were more effective than inulin in increasing populations of bifidobacteria, lactobacilli and lactococci [75].

### 3.4. Immunomodulatory Effects

Commensal BAL have several beneficial roles in the GIT including contributing to maintaining the integrity of the mucosal barrier and protecting against pathogens, as well as providing nutrients to eukaryotic cells. Thus, EPS molecules synthesized by LAB can adhere to intestinal epithelial cells, thereby impeding pathogen adhesion and/or stimulating the underlying immune system cells [76]. The interaction of the healthy microbiota and the mucosal immune system is crucial for the correct development and function of the immune system. This process takes place via the pattern recognition receptors (PRR) of host cells, which interact with the molecular effectors that are produced by intestinal microorganisms. The immune system differentially recognizes effectors synthesized by pathogens and commensal bacteria, including EPS, although their structures are similar, and they share mechanisms for interaction with the host cells; this phenomenon is called immune tolerance [77]. Thus, EPS produced by commensal and probiotic bacteria (including LAB) are involved in the modulation of (i) the innate immune response by their interaction with dendritic cells and macrophages, and (ii) the adaptive immune response, by stimulating the proliferation of T and natural killer cells. In both cases the modulation involves cytokine production by the immune system of the host eukaryotic cells, providing direct health-promoting benefits, such as fighting against pathogens and preventing of GIT cancer, as well as immunodeficiency-induced diseases, such as inflammatory bowel diseases [25,78].

A number of published reports on the in vitro and in vivo immunomodulatory or immunostimulatory effects of HoPS and HePS from LAB have recently been reviewed and some of these are detailed below [7,14,29].

In vitro experiments, including a comparative study of EPS-producing and non-producing strains, showed that the high molecular mass HePS of the probiotic *L. casei* strain Shirota [79], *L. rhamnosus* RW-9595M [80] and *L. paraplantarum* BGCG11 [81] have a suppressive effect on the activation of human macrophages as a result of increasing IL-10 and inhibition or not increasing TNF-α, IL-1, IL-6 and IL-12. In addition, HePS and sulphated-HePS, modified from *S. thermophilus* ASCC 1275, have an immunosuppressive response. Moreover, treatment with the HePS ASCC 1275 and sulphated-HePS ASCC 1275 resulted in a decrease in the pro-/anti-inflammatory cytokines (IL-1β/IL-10, IL-6/IL-10, and TNF-α/IL-10) secretion ratios of murine RAW 264.7 macrophages stimulated with lipopolysaccharide [82]. Similarly, in vivo assays demonstrated that HePS from *L. paraplantarum* CG11 had immunosuppressive activity in a mouse peritonitis model induced by carrageenan, since the levels of the pro-inflammatory mediators IL-1β, TNF-α and iNOS decreased, and increased secretion of the anti-inflammatory IL-10 and IL-6 cytokines took place [83].

HePS have also been shown to exert immunostimulatory effects. Shin et al., [84] found that HePS from *P. pentosaceus* KFT18 stimulated NO and TNF-α, IL-6 and IL-1β production in RAW 264·7 macrophages and increased the proliferation and the production of IL-2 and IFN-γ primary splenocytes in vitro. In the same study, PE-EPS treatment resulted in an increase in the thymus and spleen lymphocyte and neutrophil count in a cyclophosphamide-induced immunosuppressed mouse model.

In vivo studies in mice have revealed that oral administration of the HePS-producing *L. mesenteroides* NTM048 stimulated Peyer’s patch cells to induce IgA production at the intestinal and systemic levels [27]. The authors reported that EPS from NTM048 was an immunostimulant that enhanced mucosal IgA production [27]. In addition, the intranasal administration of EPS-NTM048 to mice with an antigen (ovalbumin) resulted in the secretion of antigen-specific IgA and IgG in the airway mucosa and the serum, suggesting that the EPS has adjuvant activity for use with mucosal vaccination [85].

With respect to HoPS, some studies have demonstrated the ability of these polymers to modulate the immune response. Sato et al. [86] found that an α-glucan (dextran) from *L. mesenteroides* strain exerted immunostimulatory activity. The levels of the IFN-ϒ and IL-10 mRNA on murine splenocytes were increased by the stimulation. It was also demonstrated that dextran from *L. sakei* MN1 can upregulate the expression of IFN-1 and IFN-γ in trout head kidney cells [87].

Furthermore, HoPS have also shown anti-inflammatory activity. Colonization of the gut of *Lactobacillus*-free mice by the fructan-producing *L. reuteri* strain 100-23, but not by the non-HoPS producer mutant, resulted in increased proportions of regulatory T-cells marked by expression of the transcription factor Foxp3, and suppressed proinflammatory T-cell responses in the spleen [88]. In the same way, β-glucan from *P. parvulus* 2.6 was able to activate human macrophages with an anti-inflammatory response [89]. In addition, the exposure of gnotobiotic zebra fish larvae to this HoPS caused the inhibition of gene expression of the pro-inflammatory cytokines, TNF-α and IL-8. Furthermore, the protein adaptor MyD88, which mediates the activation of pro-inflammatory cytokines via NF-kB, was inhibited [90]. Furthermore, the recruitment and proliferation of the neutrophils was repressed [88]. This β-glucan had an anti-inflammatory effect, which included a reduction in IL-8, both at the level of its gene expression and its secretion, in an ex vivo model of human biopsies from patients with Crohn’s disease [91].

The immunomodulatory activities of LAB-EPS are controlled by their physicochemical properties, such as monosaccharide composition, molecular weight, water solubility, electric charges and stereochemistry (this topic will not be dealt with here since it has been recently reviewed in detail by Xu et al. [7] and Zhou et al. [29]). Currently, the general opinion is that the size and charge of polysaccharides are major factors influencing the immune effect. It seems that negatively charged EPS and/or small size molecules act as stimulators of immune cells, while neutral and large EPS have a suppressor effect. Moreover, many contradictions exist with respect to this topic in the reviewed literature. This could be due to both the lack of detailed knowledge of all EPS structures, as well as to the use of different in vivo and in vitro models for the characterization of the immunomodulatory activity of these polymers. In addition, more studies associating factors, such as monosaccharide composition, functional groups, linkage patterns, and microstructures of EPS with their immune effect, will be necessary to fully understand the structure-immunity relationship of LAB-EPS [7,29].

### 3.5. Antiviral Effects

It has been proposed that the antiviral activity exerted by probiotics against diverse human and animal viruses may be mediated by mechanisms including the production of inhibitory antiviral compounds, stimulation of the immune system, and/or direct interaction with viruses [92,93]. In particular, various EPS from probiotic LAB could have antiviral effects. Based on the mechanisms proposed, these effects could be considered as (i) local or direct, where the EPS may prevent viral infection by blockage of viral adsorption by interaction with either the virus particles or the host cell [94], or (ii) systemic or indirect, as these polymers may indirectly hinder the virus by stimulating the innate and adaptive immunity of the host cell [95].

This section will discuss the studies where it has been demonstrated that the antiviral effect of EPS is totally, or at least partially, due to the action of the immune system.

It has been shown that EPS produced by LAB induce beneficial modulation of the systemic and mucosal antiviral responses and, in consequence, contribute to reducing the severity of viral infections.

In vivo oral administration of yogurt fermented with the HePS-producing *L. delbrueckii* OLL1073R-1 and the purified HePS resulted in a significant reduction in influenza virus titer and a large increase in anti-influenza virus antibodies (IgA, IgG1). Furthermore, in both groups of treated mice, the activity of natural killer (NK) from splenocytes increased significantly [96]. In porcine intestinal epithelial cells (IECs), the innate immune response, triggered by Toll-like receptor 3 (TLR3) activation, was differentially modulated by HePS from *L. delbrueckii* OLL1073R-1. EPS treatment induced porcine IECs, improving antiviral activity by TLR3 activated with poly (I:C) (synthetic analogue of viral ds RNA), which significantly increased expression of IFN-α, IFN-β, as well as the antiviral factors Myxovirus resistance A (MxA) and RNase L genes. The EPS treatment provoked reduction in the expression of IL-6 and pro-inflammatory chemokines [97]. In another in vitro study, *L. delbrueckii* TUA4408L and its HePS were able to potentiate the resistance of porcine IECs to rotavirus infection by reducing viral replication and regulating inflammatory response. It was demonstrated that the TUA4408L strain and its EPS differentially modulated the antiviral innate immune response through activation of TLR3. *L. delbrueckii* TUA4408L and its HePS were able to stimulate the signaling pathway of the interferon regulatory factor (IRF-3) and the nuclear factor-kB (NF-kB), which boosts the immune response through increasing the expression of the antiviral factor interferon (IFN)-β, MxA and RNase L [98]. Similar results were found by Mizuno et al. [99], demonstrating that the EPS of *S. thermophilus* ST538 was able to modulate the innate antiviral immune response triggered by the activation of TLR3 in porcine IECs. Furthermore, they confirmed the role of EPS in the immunomodulatory effect of the ST538 EPS strain by comparison with its EPS non-producing mutant strain.

The HePS produced by *L. plantarum* LRCC5310 reduced the duration of diarrhea, limited the epithelial lesions and decreased the rotavirus replication in the intestine, and shortened the time to recovery of young mice. In vitro analyses in the murine macrophage-like RAW 264.7 and Caco-2 cell lines, after treatment of cells with *L. plantarum* LRCC5310 EPS, revealed increased levels of the anti-inflammatory cytokine, IL-10, together with decreased levels of the proinflammatory cytokines, IL-1β and TNF-α [43].

With respect to HoPS, the purified dextrans from *L. sakei* MN1 and *L. mesenteroides* RTF10 have shown functional activity against salmonid viruses, both infectious pancreatic necrosis virus (IPNV) and infectious hematopoietic necrosis virus (IHNV). Although, the mechanism of action of the dextrans is unknown, it is feasible that they may act as immunostimulants, since the results revealed that the in vivo treatment of trout with the MN1 polymer, in addition to decreasing mortality provoked by both IPNV and IHNV, significantly increased the expression of IFN-1 (innate response) and IFNϒ (adaptive response) [87].

There are few data regarding the structural factors involved in antiviral effects mediated by the immune response. Oral administration of mice with neutral or acidic EPS from *L. delbrueckii* OLL1073R-1, prior to intranasal infection with influenza virus, was performed. The treatment with the acidic EPS, but not the neutral one, prolonged survival of the mice [96]. However, other studies demonstrated that the immunomodulatory effect from EPS-OLL1073R-1 was not similar to that of their purified APS or NPS fractions. In fact, the results indicated that the complete EPS molecule was necessary to obtain the highest immunomodulatory/antiviral activity in porcine IECs [97]. On the other hand, both APS-EPS and NPS-EPS fractions from *L. delbrueckii* TUA4408L strain were able to differentially activate the immune response, although the APS fraction was involved in the modulation of antiviral immunity [98]. The APS from TUA4408L and the EPS from OLL1037R-1, acting through recognition by different receptors, TLR4 and TLR2 respectively, induced almost identical effects on innate antiviral immunity [98]. Presumably the reason why the same response was activated by different receptors was due to a structural factor.

The immune modulating activity involves interactions at the molecular level, with the process of combining EPS to enzymes and signals depending on an extremely structural stereospecificity. However, to date, few studies have associated factors, such as monosaccharide composition, functional groups, linkage patterns, and microstructures of EPS with their immune effects [29]. Future studies, both in vitro and in vivo, will be necessary to determine the structural factors, as well as to elucidate the underlying molecular mechanisms involved in the antiviral immune response. It is important to have a thorough knowledge of the structural data of the bioactive polymers to be studied, as the isolation procedures commonly used could co-precipitate the product with other bacterial components, such as lipoproteins or lipoteichoic acid, which could stimulate an immune response [9].

Nevertheless, there is sufficient scientific evidence to propose these polymers as possible candidates in vaccine preparations; the EPS could be used as carriers of antigens, or as antigens themselves in antiviral therapy to prevent or treat viral infections in both humans and animals [24].

### 3.6. Antioxidant Effects

Reactive oxygen species (ROS) play important roles in apoptosis, cell signaling, gene expression, and ion transportation [100]. However, the accumulation of excessive ROS leads to oxidative stress that causes damage to DNA, RNA, proteins, and lipids, which can induce many diseases, including cancer and inflammation.

Currently, the use of natural antioxidants from food sources has potential because synthetic antioxidants have been restricted due to their toxic and carcinogenic effects [101]. In this sense, there are numerous studies supporting the position that some LAB-EPS can be used as effective natural antioxidants to prevent oxidative stress provoked by free radicals or ROS. The majority of these reports concern in vitro assays, with HePS having been more characterized than HoPS.

The in vitro antioxidant properties of EPS have been evaluated based on their ability to scavenge the DPPH radical, hydroxyl radicals, the ABTS radical, superoxide radicals, to inhibit lipid peroxidation and their reducing power. Oxidative damage cell models, including PC12, Caco-2, RAW264.7, induced by H_2_O_2_ have been used to evaluate the stimulation of antioxidant enzymes, such as catalase and superoxide dismutase, as well as the leakage of lactate dehydrogenase and the content of malondialdehyde [7].

Various HePS from *L. plantarum* strains have been reported to have good antioxidant activity in vitro. Some scientific studies have indicated that EPS possess free radical scavenging abilities against hydroxyl, ABTS, DPPH and superoxide radicals and strong ferrous ion chelating activity [55,102,103,104]. Antioxidant effects also involve up-regulation of both enzymatic and non-enzymatic antioxidant activities and reduction of lipid peroxidation [55,105,106]. With regards to HoPS, Du et al. [107] reported that dextran, produced by *L. pseudomesenteroides* DRP-5, has moderate DPPH radical, hydroxyl radical, superoxide anion radical, ABTS radical, and Fe^2+^ scavenging activities and reducing power.

Moreover, there are also reports of in vivo antioxidant properties of LAB-EPS. Recently, Zhang et al. [108] demonstrated that HePS from *Companilactobacillus kimchii* SR8 (formerly *Lactobacillus kimchii*) possess excellent anti-aging ability. The authors reported increased catalase and superoxide dismutase activities and decreased malondialdehyde levels in both serum and liver in a D-galactose-induced aged-mouse lipid peroxidation [55,104,105].

The reduction in the levels of malondialdehyde suggested that the intake of EPS-SR8 could oppose the lipid peroxidation in aging mice. Similar results have been found for HePS from *L. lactis* subsp. *lactis* [109] and *L. delbrueckii* subsp. *bulgaricus* B3 [110] which attenuated oxidative stress in a D-galactose-induced oxidative stress model and a rat colitis model.

The EPS antioxidant activity could be influenced by various factors, such as monosaccharide composition, molecular weight, or functional groups. In addition, the extraction and purification methods used could play a role. Among the EPS with low molecular mass, the acidic polymers often showed stronger antioxidant activities than the neutral ones [111].

Good antioxidant activity is related to the presence of hydroxyl groups and other functional groups of the EPS that produce a more stable free radical. Negatively charged groups could generate an acidic environment, which would facilitate EPS hydrolysis, exposing more hemiacetal hydroxyl groups with excellent antioxidant activity. Qin et al. found that substituted groups, such as sulfate, acetyl and phosphate enhanced in vitro the antioxidant activity of polysaccharide [112]. In this sense, it was reported that sulfonation or phosphorylation of HePS produced, respectively, by *L. plantarum* [113] and *L. lactis* subsp. *lactis* [109], increased antioxidant activity of their EPS.

The antioxidant property has also been found to be associated with chain length. In addition, since the antioxidant activity of the EPS could be determined by the number of hemiacetal hydroxyl groups, the lower the molecular mass is, the fewer the exposed hemiacetal hydroxyl groups are at equal mass concentration. Consequently, the antioxidant activity increases with decreasing molecular weight, as demonstrated in in vivo and in vitro tests [29].

The studies suggest that EPS from LAB not only act as electron donors to directly react with free radicals, or by chelating with transition metal ion catalysis, but also perform antioxidant activity by other mechanisms which need further exploration.

### 3.7. Anti-Cancer Activity

The antiproliferative nature of certain LAB-EPS has been analyzed in recent studies, highlighting potential applications for their anti-cancer effects; however, the results are preliminary. LAB-EPS usually have low cytotoxicity and side effects; for this reason, these polymers could serve as good substitutes for synthetic antitumor agents.

The possible mechanisms of anti-cancer activity exerted by EPS are as follows: (i) prevention of tumorigenesis as they could have antioxidant activity and/or by binding of genotoxic carcinogens, (ii) induction of cancer cell apoptosis, and (iii) improvement of immunity [25]. Oxidative stress and damage seem to play a crucial role in cell transformation and cancer pathogenesis. In this sense, the antiproliferative potential of EPS may be related to their antioxidant activity [6]. Several studies correlated the highest capability of EPS to inhibit tumor cells with the highest antioxidant activity. It was reported that both HePS fractions produced by *L. plantarum* 70810 [114] and *L. helveticus* MB2-1 [115,116] had high antioxidant activity with respect to hydroxyl and DPPH radical scavenging and reducing power assays. Moreover, they showed antitumor activity against Caco-2, BGC-823 and HT-29 cells. The HePS synthesized by *L. rhamnosus* SHA111 had good antioxidant activity with high values for scavenging activity towards hydroxyl and superoxide radicals and reducing power, as well as high antitumor activity against human colon cancer Caco-2 cells [32]. Antimutagenic activity (binding ability to different mutagens, such as heterocyclic amines, 4-nitroquinoline-N-oxide and 2-nitrofluorine) from EPS-producing lactobacilli (*L. plantarum* and *L. rhamnosus*) was also reported [117,118].

Apoptosis, or programmed cell death, is necessary in the treatment of cancer. In this sense, the HePS isolated from *L. acidophilus* 606 [119] and *L. fermentun* YL-11 [120] caused the early death of cancer cells, partly through the induction of apoptosis. It was reported that the HePS from *L. plantarum* NCU116 induced apoptosis of CT26 cells via TLR2 and Fas/Fasl signaling pathways [29]. Di et al. [121] found that the acidic EPS produced by *L. casei* SB27 could significantly inhibit the growth of cancer cells via caspase-3-dependent apoptosis.

Other mechanisms of antitumor action of EPS include the stimulation of cell-mediated immune responses, such as natural killer cell tumoricidal activity, T-lymphocyte proliferation and mononuclear cell phagocytic capacity [122].

Wu et al., [123] found that the EPS from *L. lactis* subsp. *lactis* promoted the apoptosis of MCF17 cells accompanied by nuclear condensation and cell shrinkage, increasing intracellular calcium levels and inflammatory cytokine production. El-Debb et al., reported that the HePS produced by *L. acidophilus* 20079 exhibited antitumor activity through both apoptotic and NF-κB inflammatory pathways [124].

In vivo assays of the antitumoral activity and its mechanisms should also be carried out to further understand the detailed functions of the EPS from LAB.

As with other biological activities exerted by LAB-EPS, the determination of the chemical composition and structure of EPS must be taken into consideration when predicting potential applications for these polymers as anti-cancer therapy. The anti-cancer activity of EPS can be affected by their physicochemical properties, such as monosaccharide composition, and the presence of β-type glycosidic linkages, uronic acid, and sulfate groups. In this regard, a relationship between the ability of EPS from lactobacilli to inhibit cell proliferation in HT-29 via apoptosis and its mannose and glucose composition has been reported [125]. Anti-HepG-2, anti-HT-29 and antioxidant activities were significantly enhanced after acetylated modification in EPS1 of *L. plantarum* 70810 [114].

### 3.8. Cholesterol-Lowering Activity

Major risk factors for cardiovascular disease include high blood pressure and elevated blood cholesterol levels. The cholesterol lowering and hypoglycemic activity of LAB-EPS have been mostly demonstrated through in vitro assays [126,127]. Some LAB-EPS seem to regulate serum cholesterol levels by inhibiting absorption of this molecule by the host [5]. In this sense, Soh et al., showed inhibition of cholesterol adsorption through an in vitro enzymatic reaction and an EPS precipitation process [128]. Although the exact mechanisms of the cholesterol-lowering effect of the EPS are still not fully understood, it has been proposed that they exert such an effect through increasing the secretion of bile acids plus behaving like dietary fibers [129]. In addition, the α-glucosidase or α-amylase inhibition activities have also been related to the hypoglycemic function of LAB-EPS [26,130]. It was found that low-fat akawi cheeses fermented with EPS-producing *L. plantarum* exhibited a better inhibitory effect on α-amylase and α-glucosidase activities than that of cheeses fermented with non-EPS producing cultures [131].

On the other hand, studies in animals and humans have also shown the effect of LAB-EPS in lowering blood cholesterol levels. The levels of cholesterol, triglycerides, and free fatty acids decreased markedly in rats fed with the kefiran of *L. kefiranofaciens* WT 2B(T). Thus, a hypocholesterolemic effect of kefiran was detected [132]. Human consumption of an oat-based food fermented with the (1,3)(1,2)-β-d-glucans *P. parvulus* 2.6 R resulted in a decrease in serum cholesterol levels, boosting the effect previously demonstrated for (1,3)(1,4)-β-d-glucans of oat-based products [133]. Kefiran-producing lactobacilli prevented the onset and development of atherosclerosis in hypercholesterolemic rabbits fed a diet containing 1% kefiran [134]. In another study, it was shown that dietary intervention with EPS-producing probiotic LAB resulted in modulation of lipid metabolism in a mouse model of atherosclerosis by reducing serum cholesterol and triglyceride (TG) concentrations [135].

## 4. The EPS of Probiotic LAB

Many probiotic bacteria (or with potential as such), are producers of a wide range of EPS. Most of them are lactobacilli strains (*L. acidophilus*, *L. plantarum*, *L. paraplantarum*, *L. casei*, *L. reuteri*, *L. paracasei*, *L. rhamnosus*, *Limosilactobacillus fermentum* (formerly *L. fermentum*), *L. gasseri*, *L. delbrueckii* subsp. *bulgaricus*, *Lactobacillus bulgaricus*, *L. helveticus*, and *L. johnsonii*). Strains belonging to *Streptococcus* spp. (*Streptococcus phocae*, *S. thermophilus*, *Streptococcus salivarius*, *Streptococcus sobrinus*), *Leuconostoc* spp. (*L. citreum*, *L. mesenteroides*, *L. mesenteroides*, *L. cremoris*, *L. lactis*), *Pediococcus* spp. (*P. parvulus*, *Pediococcus acidilactici*, *Pediococcus pentosaceus*), *Weisella* spp. (*W. confusa*, *W. cibaria*), *Enterococcus* spp. (*Enterococcus faecium*, *Enterococcus faecalis*) and *L. lactis* subsp *cremoris* have also been reported. Detailed information on these EPS-producing strains, their origin, the type of polymers they produce, and the biological functions attributed to them, has been widely reported in the literature [7,9,25,29,76].

Scientific evidence supports that some of the proposed health effects attributed to LAB probiotics can be due to the production of EPS.

A wide variety of such effects, considered as a crucial criterion for probiotic assessment, have been characterized in EPS-producing probiotic LAB, including immunomodulation, pathogen protection capability and microbiome modulation [9]. These and other local and systemic EPS actions of LAB on the host have been discussed above.

The bacterial resistance to and persistence and/or colonization in the GIT environment is highly relevant to the probiotic field. It has been demonstrated that the involvement of EPS could be of particular importance. The EPS might form a protective layer on the producing bacteria, improving their tolerance against acidic and enzymatic stress, as well as to bile salts and pancreatic juices at the duodenal level, therefore increasing their survival in the intestinal tract. Some studies have reported that some EPS have a beneficial influence on the resistance of their producing bacteria to GIT stresses [9,136], whereas other studies do not show a positive influence for EPS [81,137].

The mannose-rich HePS of *L. mucosae* DPC 6426 provides a selective advantage to this strain during gastric transit in terms of stability and persistence in comparison with non-EPS-producing strains [135]. Another study showed that the reduction in both the cell-surface-covering dextran and the HePS (galactose and glucose) produced by *L. johnsonii* FI9785 resulted in a reduction in the ability to survive GIT stresses [50]. Moreover, the resistance to GIT and technological stresses of the intestinal bacteria *L. paracasei* NFBC 338 was enhanced by the heterologous production of a β-glucan [138]. The same effect of this EPS was observed when it was evaluated using an in vivo mouse model, enhancing the survival of parental *P. parvulus* 2.6 and its isogenic non-ropy 2.6NR strains through the GIT tract [139]. In another study the viability of both strains was improved in the presence of the β-glucan using an in vivo zebra fish model [90]. Moreover, the same pediococcal β-glucan included in an oat matrix provided WCFS1 resistance to bile salts and pancreatin to the probiotic *L. plantarum* [90,140].

In contrast, no effect was observed of the β-glucan when *P. parvulus* 2.6 and 2.6NR strains were subjected in vitro to GIT stress [71]. In the same way, the glucose-rich HePS present on the surface of the producer strain *L. paraplantarum* BGCG11 was not effective enough to increase its survival under the in vitro gastric conditions assayed in comparison with the non-ropy derivatives [81].

Other studies have shown that EPS-producing strains with probiotic traits of different species of lactobacilli (*L. plantarum*, *L. fermentum*, *Lactobacillus crispatus*, *L. reuteri*, *L. pentosus*), as well as dextran-levan producer *Leuconostoc* spp. (*L. citreum*, *L. mesenteroides*, *L. pseudomesenteroides*) and *P. pentosaceus* strains, have the ability to withstand simulated human GIT tract conditions, but the direct influence of the EPS on this property has not been confirmed [63,141].

Finally, the ability of EPS attached to LAB to deal with the harsh GIT stresses could also be due to the specific ability of these strains to synthesize different EPS. For instance, Sönmez et al. [142] detected a strain-specific relationship for a protective role of EPS against gastric conditions and bile salts stress conditions in a study with 20 lactobacilli strains (*L. rhamnosus*, *L. fermentum* and *Levilactobacillus brevis* (formerly *Lactobacillus brevis*)).

Once the EPS-producing LAB arrive to the colon alive, the EPS are implicated in the interaction of the bacterial cells with the intestinal mucosa in different ways. The ability of the probiotic LAB to bind to the intestinal epithelium contributes to their persistence in the gut, enabling their beneficial effects. Therefore, one of the main criteria for the selection of probiotic strains is their ability to adhere to the intestinal epithelial cells or the mucus layer which covers this epithelium [143].

Some scientific studies suggest that in vitro EPS synthesis by probiotic LAB decreases the adhesion of the producing bacteria to enterocytes. This is the case for the high molecular weight galactose-rich HePS of *L. rhamnosus* GG [144,145] and both the two EPS (a dextran and a HePS) that cover the surface of *L. johnsonii* FI9785 [50,146] or the surface HePS of *L. plantarum* LP90 [147], as well as the dextran produced by *L. sakei* MN1 and *L. mesenteroides* strains [148,149]. In a comprehensive review regarding adhesion properties in *lactobacilli*, Castro-Bravo et al. [9] state that EPS with high molecular mass surrounding the bacterial cells might reduce or impair bacterial adhesion to intestinal cells and to abiotic surfaces due to the shielding of macromolecules which act as adhesins.

In contrast, Živkovi’c et al. [46] detected a positive influence of the HePS produced by *L. paracasei* subsp. *paracasei* BGSJ2-8 in bacterial binding to epithelial intestinal cells. The same behavior was observed for the dextran produced by *L. lactis* AV1 [150]. Other studies, both in vitro and in vivo, have shown that the presence of β-glucan enhances adhesion of *P. parvulus* strains to enterocytes [71,90,151]. Moreover, it has been demonstrated that the presence of this EPS in vivo promotes the ability of *P. parvulus* 2.6 to colonize the zebra fish intestinal tract [90]. Furthermore, Walter et al. [152] using a *Lactobacillus*-free mouse model, showed that both glucan and fructan contribute to the ecological performance of the EPS-producing *L. reuteri* TMW 1.106 during gut colonization. In contrast, the authors showed that the levan from *L. reuteri* LTH5448 did not affect the colonization of the murine GIT tract, whereas the dextran from *L. sakei* MN1, decreased the colonization capability of the bacteria in the zebrafish gut [148].

In vivo colonization and persistence of EPS-producing probiotic strains, including *L. plantarum* Lp90, have been reported in vivo using gnotobiotic zebrafish models [153]. Similarly, Lebeer et al. [144] have reported that *L. rhamnosus* GG had a higher persistence in the murine gut than its isogenic non-EPS-producing mutant, although the latter had shown good in vitro adhesion ability in a previous study [154].

Moreover, it has been shown that levan production can enhance the colonizing capability of *L. reuteri* 100-23 in *Lactobacillus*-free mice gut, accompanied by a notable induction of regulatory T-cells, which could result in immunological tolerance towards the strain [88,155].

Another beneficial property evaluated in probiotic bacteria is biofilm formation, since this characteristic could enhance colonization and increase permanence in the host mucosa [156]. The capacity of EPS-producing LAB (mostly lactobacilli) to form biofilms on abiotic surfaces has been studied, though with conflicting results regarding EPS involvement. Indeed, HePS reduction produced by *L. rhamnosus* GG [154], or both dextran and HePS from *L. johnsonii* FI9785 [50], as well as dextran from *L. sakei* MN1 [148], resulted in a decrease in their ability to form biofilms. However, a positive contribution was reported for β-glucan synthesized by *P. parvulus* and *Oenococcus oeni* strains [157] and from glucan and fructan of *L. reuteri* TMW 1.106 [152], as well as for dextran from *L lactis* AV1 [150] on their capability to form a biofilm in vitro. Meanwhile, this ability was not affected by levan production from *L. reuteri* strains, even when it was compared in vivo to the EPS-producing *L. reuteri* 100-23 and its levan-non-producing mutant strain. The authors stated that the matrix seen in electron micrographs of the fore-stomach biofilms was not formed of levan (even though, as mentioned above, levan production improved the colonization of the wild-type strain) [88,152].

It has also been demonstrated that biofilm formation could be involved in some of the postbiotic effects of the EPS synthesized by probiotics, such as protection of the host against injury produced by pathogens or their toxins [9]. Moreover, the EPS produced by some biofilm-forming LAB can impair the biofilms formation by certain pathogenic bacteria [156].

Although the above-mentioned role of EPS in all these probiotic effects are derived from different models, the differences observed may be mainly due to the diverse structures and compositions of the EPS produced by LAB. In this regard, many structural factors, intrinsic to each polymer type, such as monosaccharide composition, glycosidic bond, presence of side chains, molecular mass or charge, were reported to affect their biological, technological and protective abilities [7,67,158]. However, up to now, neither the mechanism by which the EPS are implicated in such effects, nor the exact structure/activity relationships have been completely explained.

## 5. Application of EPS-Producing Probiotic LAB in the Diet of Farm Animals

Although the previously mentioned beneficial biological functions of EPS are very interesting, both from the point of view of health and animal production, supplementation with EPS-producing probiotic LAB in the diet of food-producing animals is not a common practice (or, at least, it is not reported), other than a significant number of mono- and multi-strains inocula which have been studied in farm animals (Table 1).

However, some of these reported probiotic lactobacilli strains, *L. reuteri*, *L. amylovorus*, and *L. johnsonii*, were selected, among other probiotic characteristics, for their high capacity to produce EPS, and they were used to analyze their effect on feeding of sows and weaned piglets. The results obtained indicated that these LAB contributed to improvement in the reproductive performance of sows and weaned piglet growth, presumably due in both to a positive effect on antioxidant enzyme activity and in the immune indexes (Table 1) [164]. Although the authors did not directly correlate the biological activities to the production of EPS, it could be due to the synthesis of these polymers, as discussed in the previous section, and as recently reported for the EPS from *L. rhamnosus* GG, which has been shown to be an effective drug to relieve oxidative stress [177]. On the other hand, in vitro assays demonstrated that the lactobacilli mentioned above showed good inhibition against enterotoxigenic *Escherichia coli* (ETEC), which expressed the fimbriae K88. Similarly, other authors have reported that *L. reuteri* strains produce reuteran or levan, EPS that inhibit the ETEC binding to the mucosa, and carried out an in vivo study of feed fermentation. They concluded that the *L. reuteri* feed fermentation decreased the colonization of weaning piglets by the ETEC and that the reuteran provided by the LAB may contribute to preventing ETEC adhesion to the intestinal mucosa [178]. Later, an in vitro study demonstrated that glucan polymers from *L. reuteri* were involved in suppression of the inflammatory response to ETEC infection. This study treated porcine epithelial cells with the glucan produced by *L. reuteri*, and the global response at the level of regulation of gene expression was analyzed by high-throughput RNA-sequencing [179]. Moreover, although further in vivo studies are needed, in vitro antiviral activity was demonstrated for *L. delbrueckii* TUA4408L and its HePS, which was able to enhance the resistance of porcine intestinal epithelial cells to rotavirus infection by modulation of the innate antiviral response and reduction in viral replication [97]. Recently, EPS from *L. plantarum* prevented the adsorption of porcine epidemic diarrhea virus, and diminished injured Vero cells induced early apoptosis, as well as inflammatory responses [180].

Kefir grains are a reservoir of probiotic LAB, and they have been used experimentally in slaughter animals. In the microbiota of the kefir grains, the most representative species are *Lactobacillus kefiranofaciens*, *Lentilactobacillus kefiri* and *Lentilactobacillus parakefiri*. However, other LAB are also present (*L. paracasei*, *Lactobacillus acidophilus*, *L. delbrueckii* subsp. *bulgaricus*, *L. plantarum* and *L. lactis*) [181,182]. Thus, Bengoa et al. [181] have recently reviewed the influence of the EPS of *L. paracasei* on the probiotic properties of this BAL. Due to the instability of the microorganisms that make up the grain, their characterization over time showed variable results. Both extrinsic and intrinsic factors that affect microbial growth have a direct influence on the microorganisms that make up the grain, which is why they are usually classified as competitive exclusion cultures (CEC), a concept previously described by Nurmi and Rantala [183]. The principle of this strategy is based on the competition between the normal microbiota, the non-pathogenic intestinal microbiota and the pathogenic bacteria to colonize the GIT of the host [184]. CEC requires the addition of a non-pathogenic bacterial culture to the intestinal tract of slaughter animals to decrease colonization by populations of pathogenic bacteria [185]. CEC have been shown to be an effective method for the control of salmonellosis in commercial poultry [186]. Depending on the stage of production (state of gut maturity), the objective of CEC may be the exclusion of pathogens from the gut of the newborn animals, or the displacement of an already established population of pathogenic bacteria [187]. A novel approach is to use this tool to reduce the intestinal burdens of a bacterium that is a normal host inhabitant but pathogenic to humans, such as thermotolerant *Campylobacter* in poultry. Kefir contains *Lactobacillus* spp. and yeasts and may be a useful CEC candidate to improve the intestinal microbiota of poultry and the hygiene of the carcasses obtained from them [188]. The beneficial effect can be attributed to the inhibition of pathogenic microorganisms by metabolites (e.g., organic acids secreted by LAB belonging to the kefir grain microbiota). It has also been reported that the S-layer protein of *Lactobacillus* kefir could have a protective role against *Salmonella enteritidis* [189]. Some in vivo studies on the use of kefir in slaughter animals have been reported with encouraging results [190,191,192]. However, the beneficial effect indicators, related to both growth performance and animal health, should continue to be studied to reach more robust conclusions. The limitations and requirements of trials with big animals that must be fed with fresh probiotic products in their complex production systems may explain the lack of reports in the literature. However, the performance of such tests should be stimulated because the administration of kefir to young calves could generate a beneficial effect on production and health rates and on the immune system of animals.

There are other examples of EPS-producing probiotic LAB assayed in poultry. The *L. johnsonii* FI9785 strain was shown to be effective as a competitive exclusion agent against *Campylobacter jejuni*. A reduction in the colonization of *C. jejuni* was observed and the composition of the intestinal microbiota was significantly altered [193]. In a previous study, the structure and biosynthesis of two EPS (dextran and HePS) from this probiotic strain were reported, as well as the probable role of these polymers in biofilm formation, and host colonization as protection against the hostile environment of the intestine [48,50].

The probiotic strain *L. gasseri* F4 isolated from the GIT of free-range chickens produced a HePS. The antioxidant, antibacterial and antibiofilm activities, tested in vitro, were attributed to that polymer, so the researchers suggested its potential use as a food additive [35].

Another study demonstrated the production of EPS in four strains of *L. salivarius* isolated from chicken feces (HoPS and also HePs in one of them). These strains produced different amounts of EPS, which in turn affected the in vitro capacities of biofilm formation, autoaggregation and adhesion. The authors highlighted that all strains exhibited inhibitory effects against chicken pathogens [194].

Thus, the use of EPS-producing probiotic LAB in the diet of food-producing animals has great potential to prevent or reduce the spread of pathogens during the primary production stage and could be an effective alternative tool to the use of antimicrobials in primary food production.

## 6. Applications of EPS for the Improvement of Technological Properties of Food

Fermentation with EPS-producing LAB is used particularly in the food industry since the in situ production contributes to improving the organoleptic quality, sensory and rheological properties, as well as the stability, of the final products. Furthermore, the in situ production of EPS has the advantage of reducing the quantity of stabilizing agents used to improve textural properties in dairy and fermented cereal products. Recently, both in situ production and incorporation into foods has led to them being considered as functional ingredients, due in part to their postulated health benefits, as detailed above, and functionality in the food itself [12,13].

In general, EPS are used in the food industry mainly as: (a) viscosity agents, (b) texture, mouth feel, and freeze-thaw stability enhancers, (c) thickeners and softeners, (d) salad dressings, and (e) films and coating agents, to mention a few [130,195]. These applications of EPS are due to their physical and rheological properties which remain stable under different industrial processing conditions [52,63]. Furthermore, certain ecological characteristics, such as their biodegradability and non-toxicity, have increased their applications as natural emulsifiers compared to chemically synthesized products, which are commonly associated with a negative impact on the environment [196]. Nevertheless, it has been suggested that many of these properties depend strongly on the amount of EPS present in food products, their chemical structure, functional groups and nature of glycosidic bonds which may interact in different ways with the food matrix affecting the physical, rheological and textural characteristic of the final product [14,197,198].

### 6.1. Dairy Products

In general, EPS have been employed by the dairy industry to reduce the percentage of added milk solids, since they can be used as thickeners and stabilizers that improve the structure and consistency of foods, without modifying organoleptic properties [199]. Furthermore, EPS addition avoids the syneresis of fermented milk products, such as yogurt and cheese, even upon product storage. In the case of yoghurt, the common starter cultures *Lactobacillus delbrueckii* subsp. *bulgaricus* and *S. thermophilus* have been selected due to their ability to produce HePS in the range of 60–150 mg/L and 30 mg/L, respectively [200,201]. Recently, new possible starters or adjuvants have been studied to be applied in yogurt and cheese manufacturing.

An alternative, to counteract the problems associated with low fat content in fermented dairy products, is the use of starters with high in situ EPS production, as has been described for the HePS producer *Limosilactobacillus mucosae* DPC 6426 (formerly *Lactobacillus mucosae*), which improved the functional and rheological properties during the production of a low-fat yoghurt, as well as significantly decreasing syneresis [135]. In the same manner, low-fat content impacts on the texture and development of the flavor of cheese [14,202], and, as in the case of yogurt, an EPS-producer added as starter or adjuvant culture, has been shown to improve the texture and quality of fat-reduced cheeses [200]. In low-fat mozzarella, and semi-fat or low-fat cheddar cheese, the addition of some HePS-producing strains of the genus *Lactobacillus (L. delbrieckii* subsp. *bulgaricus* MR-1R), in combination with *S. thermophilus* and *Lactococcus lactis* ssp. *cremoris* DPC6532, increased moisture content and yield, while improving its melting, but had no negative impact on flavor, when compared with control cheeses [203,204].

A study of different EPS produced by different strains of *L. lactis*, in a model of a low-fat fresh cheese, revealed that, in different ways, they modified the structural and macromolecular properties of the product. In this model, one strain, which produced a ropy capsular EPS (LL-1+), provided higher gel stiffness, increased water retention, lower particle size and improved creaminess texture, in comparison to a non-ropy EPS (LL-2)-producing strain [205]. Nguyen et al. [19] reported that, to avoid some quality defects in yogurt texture (graininess), certain criteria must be considered, such as the morphology of the bacteria and the structure of the EPS produced by them, as well as the final EPS concentration and the rate of acidification in the product.

In Scandinavian fermented milk drinks, such as viili, taette, fil, and skyr, the rheological characteristics, such as firmness, thickness and sliminess, rely on the ability of ropy strains of *L. lactis* subsp. *lactis* and *L. lactis* subsp. *cremoris* to produce different HePS that confer texture to the final product [206].

For the production of ice-cream at industrial scale, indispensable additives for its manufacture, such as gums stabilizers, galactomannan hydrocolloids or chemically modified plant carbohydrates (starch, pectin, guar gum, etc.) and glucomannan based “salep”, are required. However, EPS have shown some functional properties as stabilizers, and it may be possible to replace them by in situ EPS production by LAB. Dertli et al. [207] analyzed the impact of in situ HePS production by *S. thermophilus* strains on different technological and sensorial properties of ice cream, in an effort to develop a functional ice-cream without adding common stabilizers. A rheological analysis of the ice-cream showed an improvement in viscosity which was associated with EPS properties as a thickening and gelling agent, while other techniques demonstrated the presence of a web-like compact microstructure with holes that also correlated with the improved rheological properties.

### 6.2. Meat Products

Since early times, organic acids, such as lactic acid and acetic acid, have been employed as the principal natural method of preserving raw meat. *Lactobacillus* and *Pediococcus* are the main genera used to improved food safety in raw meat, as they produce lactic and acetic acids which decrease the population of other indigenous bacteria. Bacteriocin or nutrient competition may also contribute to this safety, and the development of color and texture, as well as other characteristics of meat products [129]. In the meat industry, hydrocolloids and phosphates are the main ingredients intentionally incorporated to enhance the quality of meat products [208]. Hydrocolloids have been used to improve water holding capacity, emulsion stability, and influence the gelling properties of meat proteins, yield and juiciness of cooked products, spread ability and mouthfeel in fat-reduced products or to upgrade the textural characteristics of low or reduced fat content [119,209,210].

Nowadays, consumer demand for low-fat and/or additive-free meat products is increasing, so the development of new products in the meat industry is important in this market area. However, it can also cause drawbacks as low-fat processed meat products have some defects regarding their technological and sensory qualities [129,211]. In this context, in situ EPS production by LAB seems a promising alternative. Few studies have described analysis of this production in meat products, including cooked ham, raw fermented sausages, and spreadable raw fermented sausages, in which the polymers enhance water-binding capacity or reduce the amount of fat added into the products. This last effect will be beneficial for human health, since ingestion of high-fat meat products is associated with diabetes and cardiovascular disease [208].

Dertli et al. [207] evaluated the effect of two EPS producers (HoPS and HePS producers, respectively), *Lactiplantibacillus plantarum* 162 R (formerly *Lactobacillus plantarum)* and *Leuconostoc mesenteroides* N6 on a Turkish fermented sausage. The final products showed an improvement in their textural properties (less sticky, harder and tougher with a web-like structure in the sausage matrix) in comparison with those sausages containing a non-EPS producer. Similarly, the in situ HePS production by *L. plantarum* TMW 1.1478 was evident on the final properties of salami (a dried type of fermented sausage), which contributed negatively to the quality attributes, as it was significantly softer despite not been affected in terms of its sensorial properties [212].

Additionally, in the research by Hilbig et al. [213], the effect of two EPS was evaluated on the spread ability of a fat-reduced fermented raw sausage (“Teewurst”). *Lactilactobacillus sakei* TMW 1.411 (formerly *Lactobacillus sakei)*, as well as *Lactilactobacillus curvatus* TMW 1.1928 (formerly *Lactobacillus curvatus*), both HoPS producers, were inoculated as starter cultures in a raw sausage model. The final products revealed that EPS produced by both strains during fermentation made it possible to reduce around 20% of fat content, thereby significantly improving the softness and spread ability in those sausages with EPS compared to control samples containing non-EPS producer or inoculated with the HePS producer, *L. plantarum* TMW 1.1478, which had a negative effect on the quality of the product.

### 6.3. Fermented Beverages

The organoleptic and sensorial characteristics of some fermented beverages, such as kefir and pulque, have been attributed to LAB and to their EPS production in situ during the fermentation process [12,195].

Kefir, a self-carbonated and slightly alcoholic dairy beverage consumed in Eastern Europe, is prepared from kefir grains, which are mainly composed of proteins and polysaccharides, as well as a combination of LAB (homo- and hetero-fermentative), acetic bacteria and yeast, a symbiotic consortium characteristic of grains [182,214]. During fermentation, peptides and polysaccharides (kefiran) are produced, acting as viscosity agents. Kefiran, is a soluble glucogalactan composed of equivalent parts of glucose and galactose, which has been reported to have a molecular mass between 10^5^ and 10^7^ Da [132,215], and is mainly produced by *Lactobacillus kefiranofaciens* and other lactobacilli [216]. Other EPS-producing LAB have been isolated from kefir grains, such as *L. plantarum*, *Lacticaseibacillus paracasei* (formerly *Lactobacillus paracasei*), *Lactobacillus helveticus* (formerly *Lactobacillus suntoryeus*), *Lactiplantibacillus pentosus* (formerly *Lactobacillus pentosus*), *L. lactis* subsp. *lactis* and *L. mesenteroides* [217,218].

Pulque, is an alcoholic beverage made from the fermentation of aguamiel, the fresh sap extracted from diverse agave species, and is consumed mainly in those regions in Mexico where these species of agave are cultivated. The final sensory characteristics in pulque are determined by the fermentation time of the aguamiel, the increment in viscosity as a consequence of the production of EPS and the degree of alcohol produced [219]. *L. mesenteroides* IBT-PQ, has been recognized as one of the most important microorganisms in pulque fermentation, producing a soluble linear dextran with glucose molecules linked by α-(1,6) bonds with branching from α-(1,3) bonds synthesized from sucrose present in aguamiel and pulque. Other *Leuconostoc* species, such as *Leuconostoc citreum* and *Leuconostoc kimchi*, have also been reported as the most abundant LAB species during the early stages of pulque fermentation, so it has been proposed that different EPS production may contribute to the organoleptic characteristics of both beverages [219].

In fermented soymilk products, EPS-producing LAB act as functional starter cultures as they contribute to the consistency and rheological properties [220]. Various soybean-based products, such as soy sauce and soy paste, have recently been produced using EPS-producing LAB [198]. Li et al. [221] monitored the fermentation over 21 days of soy milk with two EPS-producing *L. plantarum* 70810 and *L. rhamnosus* 6005 starters. The fermented soy milk with *L. plantarum* 70810 conserved the viscosity profile, increased the technological properties and enhanced the mouth-taste of soy milk.

### 6.4. Cereal-Based Beverages and Food Products

Cereal-associated LAB produce a wide variety of EPS and oligosaccharides due to glycansucrase activity, enhancing both the textural and organoleptic characteristics of the final product. It is feasible to use them as probiotics and their polymers as prebiotics and/or postbiotics. In this context, Pérez-Ramos et al. [76] evaluated the ability to produce β-glucan by the *P. parvulus* 2.6 R (ropy strain) and the isogenic non-producing strain *P. parvulus* 2.6 NR (non-ropy), in three different cereal-based matrices. The levels of β-glucan were higher in oat and rice flours fermented with the ropy strain and enhanced rheological properties in the final product were observed.

Lorusso et al. [222] evaluated quinoa flour-based fermented beverages with different LAB, including the probiotic *L. rhamnosus* SP1, an EPS-producing *Weissella confusa* DSM20194, and the *L. plantarum* TB610 strain isolated from quinoa. After 20 h of fermentation, increment in the viscosity and water-holding capacity of the quinoa beverage due to a dextran-type EPS produced by *W. confusa* was observed. In addition, a stable EPS–protein network which also contributed to enhanced textural properties of the product was registered.

### 6.5. Dough and Bakery Products

The two main functions of hydrocolloids incorporated into dough systems and breads are: (i) the water holding capacity and water distribution, and (ii) structural interactions with components such as gluten, non-gluten proteins, and starch. Additionally, hydrocolloids may stabilize the interface of the dough liquid film surrounding gas bubbles, thus improving gas retention [223].

HoPS producers are usually incorporated in products such as sourdough, as they have a certain amount of sucrose, which serves as a carbon source for the in situ production of these type of EPS [206], and influences the structural quality and baking ability in the bakery. Dextran produced by bacterial fermentation and added at a level below 5% as a food ingredient in the baking industry was approved by the European Commission in 2000, on the basis that it does not constitute a hazard for consumer health [224].

The use of EPS producer starters has attracted the attention of the bakery industry, as an increase in demand for gluten-free products must be met. However, baking without gluten, the main ingredient that influences the structure and quality of bread, continues to be a challenge. In addition, cereals free of gluten are employed to produce gluten-free (GF) products. Gluten-free grains include sorghum, rice, corn, millet and teff, and the pseudocereals, amaranth, quinoa and buckwheat [225]. However, the flours obtained from these grains lead to products with poor structure, texture and mouthfeel. GF breads have low water absorption, appreciable changes in crumb features, volume reduction, and poor stability of bread. These defects could be prevented by incorporation of EPS-producing LAB into gluten-free sourdough [6].

Some evidence suggests that EPS, such as dextran produced by *L. mesenteroides* and *W. cibaria*, levan from *Fructilactobacillus sanfranciscensis*, (formerly *Lactobacilllus sanfranciscensis*), and reuteran from *L. reuteri*, clearly affect dough rheology and bread texture, and it may be possible to replace or reduce the use of hydrocolloids in wheat and gluten-free baking [201,226]. The above was shown by Katina et al. [227] during the in situ dextran production by *W. confusa* VTT E-90392 in wheat sourdough, which improved the viscosity of the sourdough, and increased the final volume of the bread (around 10%) and the crumb softness (25–40%) without producing strong acidity. Similar results were reported for the in situ synthesis of dextran by *W. cibaria* MG1 in gluten-free sourdoughs (buckwheat, quinoa, and teff) and a *W. confusa* on a whole grain pearl millet bread, in which the dextran produced improved bread volume, reduced crumb firmness and avoided moisture lost [228,229].

Furthermore, due to an increased demand for unconventional flours (e.g., legumes, pseudocereal, etc.), sourdoughs prepared with unusual grains might be a good source for EPS-producing strains. Legume-base sourdough seems a potential alternative, despite the lack of gluten in flours obtained from these, which can be overcome by the in situ production of EPS during the fermentation process [230]. *W. confusa* Ck15, a dextran producer, was isolated as a dominant strain after several backsloppings of a spontaneously fermented chickpea flour dough. In situ dextran production by *W. confusa* Ck15 fermentation improved the viscosity and the production of the EPS percentage in the doughs with respect to the other doughs inoculated with another dextran producer, *Leuconostoc pseudomesenteroides* DSM 201193, and a non-EPS producer *L. plantarum* F8 [230].

## 7. Future Perspectives

The use of EPS-producing LAB and their polymers in the dairy industry is well established, and their usage in the development of meat and cereal-based food is predicted to expand in the near future. These foods will be used principally for groups of the population that require low-fat or gluten-free fermented products.

The search for new EPS producing LAB will be facilitated by the fact that many of them can be isolated from the GIT and/or animal feces, and the use of these bacteria will place emphasis on their application according to their origin. Nevertheless, concerning the isolation and purification of LAB-EPS, special care should be taken to prevent degradation of the polymers during their extraction. Special measures should also be taken to avoid the presence of any impurities, such as lipopolysaccharides, which can interfere with the characterization of the polymer’s biological activity. Furthermore, EPS structures can differ significantly between bacterial strains and result in different biological properties. Details of the specific EPS structures used in research are often lacking or are insufficient to determine the exact structure–function relationships. When reporting findings on the biological functions of EPS, special care should be taken in providing detailed information on the bacterial strain used, and the specific EPS structure.

The livestock industry is important worldwide, and the farming industry is demanding improvement of animal feed with prebiotics and postbiotics. Nevertheless, the market for feedstuff LAB is still in an early stage of development and currently the number and volume of commercialized products containing LAB is relatively small. Moreover, there are no probiotic EPS-producing LAB nor bacterial EPS currently used in animal health. Thus, this is a new field which deserves to be explored and exploited. LAB have shown beneficial effects in food-producing animals through various mechanisms. According to the background described previously, and, in particular, in farm animals, the EPS produced by LAB may be part of the mechanisms by which bacteria generate some of the beneficial effects in animals. EPS can enhance the viability of LAB in the gastrointestinal tract, both due to its resistance to gastrointestinal conditions, and by increasing its permanence in the intestine (by biofilm formation, they can increase interaction with the cells of the intestine—but in some cases they decrease it because they interact with adhesion factors). The administration of both EPS-producing strains and purified EPS can preventatively or therapeutically protect host animals from pathogen infections, increase the immune response and improve feed conversion, which results in lower morbidity and mortality, and greater feed efficiency, thereby generating economic benefits on farms. In turn, these benefits are added to the fact that the use of EPS, or strains that produce, them can replace antimicrobial drugs.

Intensive animal husbandry has increased the use of antibiotics both preventatively and therapeutically. The use of alternative agents to antibiotics should be encouraged and a wide range of tools should be available to prevent antibiotic resistance. Currently the addition of antibiotics into feedstuffs for food-producing animals is a global trend, and their use produces an increase in antimicrobial resistance, which can be transmitted throughout the food chain and generate health problems in consumers. The search for EPS-producing LAB, as well as the elucidation of the mechanisms of action of EPS in food-producing animals, will increase the quantity of EPS, or producing strains available for administration on farms, as well as the knowledge of the expected beneficial effects, and will enhance the availability of alternative strategies to the use of antimicrobials.

## Figures and Tables

**Table 1 foods-11-01284-t001:** Examples of use of LAB and their applications on animals.

Food-Producing Animals	Probiotic LAB	Probiotic Role	References
**Bovine Production**			
Dairy Cows	Lactococcal (+prebiotic)	Preventing mastitis by diminishing inflammation of mammary gland and reducing the levels of mastitis-causing pathogens (enterococci and streptococci).	[159]
Calves	Normosil: *L. brevis* B-3, *L. plantarum* 8 TIMES, *L. acidophilus* 457, *Enterococcus faecium* UDS 86,	Increasing the levels of normal microbiota, such as LAB and bifidobacteria, and decreasing the prevalence of *Escherichia coli*. Increasing red blood cells, the concentration of hemoglobin, and γ-globulins within the physiological range. Improving phagocytic reaction in the blood serum. These effects suggest a high response of calves to infectious agents.	[160]
Calves	*Lactobacillus uvarum* LUHSS245	Increasing and decreasing, respectively, levels of LAB and enterobacteria in feces. Positive influence on certain health parameters in blood; reduction in lactate and serum alanine aminotransferase (AST) concentrations.	[161]
Calves	*L.s casei* DSPV 318T, *Lactobacillus salivarius* DSPV 315T and *Pediococcus acidilactici* DSPV 006T	Earlier consumption of starter and, indirectly, presumed stimulation of earlier development of the rumen, omasum and reticulum, thus favoring early weaning. Inoculated calves had better growth performance. This behavior could be due to better digestion of lactose and spray-dried whey proteins.	[162]
Calves	*L. casei* DSPV 318T, *L. salivarius* DSPV 315T and *P. acidilactici* DSPV 006T, *L. plantarum* DSPV 354T.	The inocula were added to a computerized milk feeder system. The fermentation increased the shelf life of the milk, avoiding, in consequence, the need to frequently discard the milk with a cost reduction. The inocula stimulated milk intake. *Lactobacillus*/coliform ratio was >1 in the probiotic groups and <1 in the control group. Detection of increased gain of body weight gain and feed efficiency.	[163]
**Swine Production**			
Sows and Piglets	*L. reuteri*, *L. amylovorus*, and *L. johnsonii*	Sows Improvement in reproductive performance including increased number of birth piglets and birth weight per litter, conception rate during estrus, and lower numbers of weak piglets. Increased antioxidant capacity—concentration of malondialdehyde activity (MDA) was lower in the group fed with the LAB. Increasing immune indexes—TNF-and IgA were higher compared with the control. Piglets Improvement of growth development and decrease in diarrhea incidence—LAB increased final body weight and decreased incidence of diarrhea. Increase in average daily gain, daily feed intake, and the efficiency of feed utilization of piglets in the group fed with the LAB. Increased antioxidant activity—concentrations of superoxide dismutase (T-SOD) were significantly higher in the LAB-treated group and the MDA concentration was lower. Increased immune indexes—TNF, IgA, IgG were higher in the group fed with LAB than the control.	[164]
Piglets	*L. salivarius*	The inocula increased the number of lactobacilli and villus height in the duodenum, jejunum, and ileum.	[165]
**Poultry**			
Poultry	*Enterococcus faecium* LET 301, *L. salivarius* LET 201, *L. reuteri* LET 210,	Prevented negative effects of antinutritional factors, such as dietary lectins soybean agglutinin (SBA) and wheat germ agglutinin (WGA). The treatment with the bacteria provoked an increase in several digestive enzyme activities and alkaline phosphatase (an intestinal maturation marker) in birds fed with a conventional diet. In addition, supplementation with the bacteria counteracted: (i) the deleterious effects of SBA increased content of SBA, and (ii) the negative effect of a WGA dietary source on the activity of digestive enzymes and intestinal mucosa integrity.	[166]
Eggs	*L. lactis* subsp. *cremoris*	In the spleen and cecal tonsils of broiler chickens, prebiotics or synbiotics produced by the BAL stimulated gene expression involved in energy metabolism and immune response.	[167]
Broiler	*L. salivarius* and *P. parvulus*	Improvement in immune response, bone characteristics, weight gain and intestinal morphology, as well as decrease in *Salmonella enteritidis* colonization.	[168]
Broilers (eggs)	*Bifidobacterium bifidum*, *G3; B. animalis*, *G4; B. longum*, *G5; or B. infantis*, *G6*	Treatment with bifidobacterial, instead of LAB, increased broiler growth performance detected by body weight, weight gain and ratio of feed conversion. Enhanced thyroid hormone metabolism since concentrations in serum of thyroxin and triiodothyronine were elevated upon bacterial treatment. Ileal architecture was improved: higher villus height values and the villus height/crypt depth ratio. The levels of ileal LAB and *Bifidobacterium* spp. increased. By contrast, total coliform levels, as well as bacterial counts, decreased.	[169]
Broilers	*Lactobacillus* *Bifidobacteria*	Improved body weight gain and prevention of the deleterious and/or lethal effects of *Salmonella* infection in chicks by two mechanisms: competitive exclusion and the enhancement of cytokine secretion.	[170]
Broilers	*L. salivarius* DSPV 001P	Improved body weight and tendency to reduce the rate of mortality.	[171]
**Sheep production**			
Sheep	*L. rhamnosus*	Rumen microbiome structure and abundance were slightly altered.	[172]
**Goat production**			
Goat	*L. rhamnosus* and *E. faecalis*	Enhanced weight gain and drop in the gut pH, thus maintaining an equilibrium of ruminal microbiota	[173]
**Aquaculture**			
Juvenile convict cichlid fish	*L. casei* PB-LC39	Improved growth: final body weight, percentage of weight gain (%), specific growth rate, food conversion ratio, and protein content of whole-body composition were significantly higher. Increased activity of digestive enzymes (protease, amylase, and lipase). Improved immune response (levels of total immunoglobulin (Ig), and serum globulin were increased). Changes in the microbiota: levels of LAB in fish gut were enhanced. After an air-dive test, the rates of fish recovery in the group fed with the LAB was significantly higher than that in the control group.	[174]
Snakehead fish (*Channa argus*)	*L. lactis* L19 and *E. faecalis* W24	Increased feed efficiency ratio, specific growth rate, final body weight, weight gain, and protein efficiency ratio. Increased IgM, ACP, AKP, LZM, C3 and C4 activity in serum, which could induce humoral immunity. Up-regulation of the expression of IL-1β, IL-6, IL-10, TNF-α, IFN-γ, HSP70, HSP90, TGF-β in the spleen, head kidney, gill, liver and intestine. After challenge with *Aeromonas veronii*, increased survival rates and resistance to disease.	[175]
**Apiculture**			
Honey bee	*L. brevis* B50 Biocenol	Increased resistance to infectious diseases and stress conditions: increase in the ratio of lactic acid bacteria to enterobacteria. Enhanced immunity in bee colonies: increased expression of antimicrobial peptides (abaecin, defensin-1) coding genes and pattern recognition receptors (Toll-like receptor and peptidoglycan recognition proteins).	[176]

## Data Availability

Not applicable.

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
