# Peer review of "Biological Functions of Exopolysaccharides from Lactic Acid Bacteria and Their Potential Benefits for Humans and Farmed Animals"

_foods, 2022, doi:10.3390/foods11091284_

Round 1

Reviewer 1 Report

The topic is interesting and the authors comprehensively described an information of EPS charactersitics, role of EPS in a few groups of foods (dairy products, meat products, fermented beverages, cereal-based beverages, dough and bakery products), biological functions of EPS, and application of EPS in the animal diet. I find the presented data appropriate and I am impressed by the wide approach to the topic. Still, there are several issues raised.

The main weak point is the lack of methodological section. According to „Instructions for Authors” systematic reviews published in Foods should follow the PRISMA guidelines. I suggest that the authors have added at least informations such as:
- inclusion and exclusion criteria for the review and how studies were grouped for the syntheses
- databases, registers, websites, organisations, reference lists and other sources searched or consulted to identify studies
- search strategies for all databases, registers and websites
- methods used to decide whether a study met the inclusion criteria of the review
- methods used to decide whether a study met the inclusion criteria of the review,
- methods used to collect data from reports, including how many reviewers collected data from each report; and other. 

These are key-points considering that the presented results also refer to a role of EPS in human health.

All manuscript is very long. If the paper aims to present data on the characteristics of EPS, then comprehensive data on probiotics should be presented in the background. All the information on probiotics (lines 420-553) is known and well described in scientific papers. Please consider deleting this information or inserting a summary version in the introduction. 

Author Response

According to the reviewer suggestion, we have deleted the information concerning to the probiotics (previous lines 420-553). Moreover, we have reorganized previous sections to decrease the length of the manuscript.

Concerning to the methodology, we consider that it is not necesary to add a methodological section, because, as far as we know there are not meta-analyses published concerning to the health benefit of EPS, and this section is not included in other reviews of this kind published in FOODS.

However, we would like to detail below our approach to select the results described in the review:

  • We have utilized information taken from web sites such as:

PubMed (National Library of Medicine, National Center for Biotechnology Information) https://pubmed.ncbi.nlm.nih.gov/: Scopus (Elsevier) https://www.scopus.com/search/form.uri?display=basic; ScienceDirect
https://www.sciencedirect.com/ ; Tandfonline (Taylor & Francis Online)
https://www.tandfonline.com/; Frontiers https://www.frontiersin.org/;
Semantic Scholar https://www.semanticscholar.org/; MDPI https://www.mdpi.com/
ResearchGate https://www.researchgate.net/search; CAS Scifinder-n  https://scifinder-n-cas-org.ehu.idm.oclc.org/

  • To select the references, we have taken in account the expertize of the groups contributing to the review: (1) The groups of Dr Mª Teresa Dueñas from the Basque Country University (Spain) and Dr Paloma López from the Biological Resarch Center Margarita Salas (CSIC, spain), with an experience of more than 20 years in the discovery and characterization of EPS from LAB and (2) the group of Dr. Laureano Frizzo from National University of the Litoral-National (Argentina) expert in farm animal health.

Reviewer 2 Report

Overall, this is a relevant and well-written review, based on bibliography. The authors cite a lot of data in the manuscript (review), but I have noticed some weaknesses in terms of structure.

The manuscript sections could be reorganized to better suit the title and purpose of the manuscript.

I find a disproportionately extensive description of probiotics (3, 3.1, ....), and especially Probiotics in farm animals (3.3.2.).

3.6.4. Immunomodulatory effects replace  by 3.5.4 Immunomodulatory effects replace

Author Response

According to suggestion of the reviewer we have decreased the length of the paper removing previous sections 3, 3.1 and 3.3.2 and corrected the typographic error (3.6.4).

Round 2

Reviewer 1 Report

The authors have adequately addressed all reviewers remarks and the improved paper can be now accepted in the present form.